# Extracorporeal Shock Wave Therapy Protected the Functional and Architectural Integrity of Rodent Urinary Bladder against Ketamine-Induced Damage

**DOI:** 10.3390/biomedicines9101391

**Published:** 2021-10-04

**Authors:** Yen-Ta Chen, Kuan-Hui Huang, John Y. Chiang, Pei-Hsun Sung, Chi-Ruei Huang, Yi-Ching Chu, Fei-Chi Chuang, Hon-Kan Yip

**Affiliations:** 1Division of Urology, Department of Surgery, Kaohsiung Chang Gung Memorial Hospital, College of Medicine, Chang Gung University, Kaohsiung 833, Taiwan; yenta1965@yahoo.com.tw; 2Center for Shockwave Medicine and Tissue Engineering, Kaohsiung Chang Gung Memorial Hospital, Kaohsiung 833, Taiwan; e12281@cgmh.org.tw (P.-H.S.); starchmay33@gmail.com (C.-R.H.); yiching08@gmail.com (Y.-C.C.); 3Department of Obstetrics and Gynecology, Kaohsiung Chang Gung Memorial Hospital, Kaohsiung 833, Taiwan; gynh2436@adm.cgmh.org.tw; 4Department of Computer Science and Engineering, National Sun Yat-Sen University, Kaohsiung 804, Taiwan; chiang@cse.nsysu.edu.tw; 5Department of Healthcare Administration and Medical Informatics, Kaohsiung Medical University, Kaohsiung 807, Taiwan; 6Division of Cardiology, Department of Internal Medicine, Kaohsiung Chang Gung Memorial Hospital, College of Medicine, Chang Gung University, Kaohsiung 833, Taiwan; 7Institute for Translational Research in Biomedicine, Kaohsiung Chang Gung Memorial Hospital, Kaohsiung 833, Taiwan

**Keywords:** extracorporeal shock wave, ketamine, urinary bladder dysfunction, inflammation, cell stress signaling, oxidative stress

## Abstract

This study tested the hypothesis that extracorporeal-shock-wave (ECSW) protected the functional and anatomical integrity of rat urinary-bladder against ketamine-induced damage. In in vitro study, the rat bladder smooth muscle cells (RBdSMCs) were categorized into G1 (sham-control), G2 (RBdSMCs + menadione), G3 (RBdSMCs + ECSW) and G4 (RBdSMCs + menadione + ECSW). The results showed protein expressions of oxidative-stress/mitochondrial-damaged biomarkers (NOX-1/NOX-2/oxidized protein/cytosolic-cytochrome-C/cyclophilin-D), inflammatory markers (MyD88/TRAF6/p-IKB-α/NF-κB/TNF-α/IL-6/IL-1ß/MMP-9/iNOS), and cell-stress response signalings (ASK1/p-MKK4/p-MKK7/ERK1/2//p-JNK/p-p38/p-53) were significantly increased in G2 than in G1 and G3, and those were significantly reversed in G4 (all *p* < 0.0001). Adult-male SD rats (*n* = 24) were equally categorized into group 1 (sham-control), group 2 (ketamine/30 mg/kg/daily i.p. injection for four weeks), group 3 [ketamine/30 mg/kg + ECSW/optimal energy (0.12 mJ/mm^2^/120 impulses/at 3 h and days 3/7/14/21/28 after ketamine administration)] and group 4 [(ketamine/30 mg/kg + ECSW/higher energy (0.16 mJ/mm^2^/120 impulses)] and animals were euthanized by day 42. The results showed the urine levels of pro-inflammatory cytokines (TNF-α/IL-6) were lowest in group 1, highest in group 2 and significantly higher in group 3 than in group 4 at days 1/7/14/28 (all *p* < 0.0001). The duration of urinary bladder contraction was lowest in group 2, highest in group 1 and significantly higher in group 4 than in group 3, whereas the maximal pressure of urinary bladder exhibited an opposite pattern of bladder contraction among the groups (all *p* < 0.0001). The histopathological findings of fibrosis/inflammation/keratinization and protein expressions of oxidative-stress/mitochondrial-damaged biomarkers (NOX-1/NOX-2/oxidized protein/cytosolic-cytochrome-C/cyclophilin-D), and inflammatory (TLR-2/TLR-4/MyD88/TRAF6/p-IKB-α/NF-κB/TNF-α/IL-1ß/MMP-9/iNOS) and cell-stress response (ASK1/p-MKK4/p-MKK7/ERK1/2//p-JNK/p-p38) signalings and apoptotic/fibrotic biomarkers (cleaved-caspas3/cleaved-PARB/Smad3/TFG-ß) exhibited an identical pattern of urine proinflammatory cytokine among the groups (all *p* < 0.0001). ECSW effectively attenuated ketamine-induced bladder damage and dysfunction.

## 1. Introduction

Ketamine, a non-competitive N-methyl-D-aspartic acid receptor antagonist, was first discovered more than sixty years ago and was used as a clinical application of anesthetic [1]. Of late, ketamine-induced lower urinary tract syndrome (LUTS) has attracted increased attention due to the rising abuse of ketamine in recent years as the role of this drug has become recreational among young adults [2,3,4,5]. Abundant data have shown that ketamine abuse (i.e., long-term ketamine abuse) commonly induced urological sequelae [6,7], including syndromes (LUTS) that bear a resemblance to interstitial cystitis [8]. Additionally, LUTS are frequently linked with reduced bladder capacity, urine incontinence, hematuria and suprapubic painful sensations that have been identified due to neurological disorders [8,9], including (1) direct toxic injury on the urothelial layer causing bladder barrier dysfunction; (2) chronic neurogenic inflammation; and (3) immunoglobulin E-mediated hypersensitivity [10]. Intriguingly, a more recent experimental study [11] has also displayed that ketamine treatment markedly increased bladder weight, high bladder/body coefficient, contractive pressure of the urinary bladder, voiding volume, dysregulated the urinary bladder components and damaged the glycosaminoglycan layer as well as reduced bladder compliance. However, the exact causative mechanistic basis underlying the association between ketamine abuse and ketamine-caused cystitis, fibrosis and LUTS is still currently unclear [12]. Of distinctive importance is that there is still lacking an effective treatment for Ketamine-induced LUTS. In particular, these patients usually require long-term diaper use which always deprives them of the ability to take a long journey. 

Our previous study [13] revealed that ECSW therapy ameliorated cyclophosphamide-induced rat acute interstitial cystitis through inhibiting inflammation and oxidative stress in both in vitro and in vivo experimental studies. Additionally, another previous study [14] of ours showed that ECSW therapy suppressed the inflammatory reaction and restored urothelial barrier integrity in acute interstitial cystitis by upregulating the fatty acid receptor GPR120. Furthermore, our recent study [15] has demonstrated that ECSW treatment effectively inhibited radiation-induced chronic cystitis, preserved the urinary bladder contractility and reduced urine retention. Intriguingly, our more recent studies have established that ECSW effectively preserved neurological function in condition of diabetic neuropathy [16] and relieved the neurological pain [17]. Based on the aforementioned studies [13,14,15,16,17], we have proposed that ECSW therapy might improve the ketamine-elicited urinary bladder dysfunction, i.e., incontinence (UI) and urinary retention (UR).

## 2. Materials and Method

### 2.1. Ethics Statement

Our animal procedure and protocol were certified by the Institutional Animal Care and Use Committee at Kaohsiung Chang Gung Memorial Hospital (Affidavit of Approval of Animal Use Protocol No. 2019032501).

### 2.2. In Vitro Study

Rat Urinary Bladder Smooth Muscle Cells (CSC-C9375W) (RBdSMCs) were purchased from Creative-Bioarray Com. and were cultured in T25 flask for expansion. The cells were divided into group A [RBdSMCs (1 × 10^6^ per mL) + vehicle], group B [RBdSMCs (1 × 10^6^ per mL) + menadione (25 μM) (i.e., menadione acted as an oxidative-stress compound) (menadione treated the cells for 30 min, followed by washing and then continuously cultured for 24 h], group C [RBdSMCs (1 × 10^6^ per mL) + ECSW (0.12 mJ/mm^2^ for 180 impulses)] which was applied to the culture disk/ECSW treatment at 3 h after cell culturing, followed by culturing for 24 h and group D [RBdSMCs (1 × 10^6^ per mL) + menadione (25 μM) + ECSW (0.12 mJ/mm^2^ for 180 impulses)]. The procedure, protocol, dosage of menadione and energy of ECSW were based on our previous reports [17,18]. Additionally, 24 h after the cell culture, the cells were collected in each group for the individual study to delineate the underlying mechanism of ECSW on inhibiting the inflammation and oxidative stress.

#### 2.2.1. Creating UR and UI Animal Model by Ketamine Administration and Animal Grouping

The procedure and protocol were based on our previous report [19] and recent report from other investigators [20] with some modification.

Experiments were performed on adult-female Sprague-Dawley rats (Animal Center of BioLASCO, Taipei, Taiwan), weighting between 250 and 275 g. Adult-male SD rats (*n* = 24) were equally categorized into group 1 [sham-control, i.e., 1.0 cc saline by daily intraperitoneal injection for 4 weeks], group 2 [ketamine (30 mg/kg) daily intraperitoneal injection for 4 weeks], group 3 [ketamine 30 mg/kg + optimal ECSW energy (0.12 mJ/mm^2^, 120 impulses applied into the pelvic surface area at 3 h and days 3, 7, 14, 21 and 28 after ketamine administration)] and group 4 [ketamine (30 mg/kg) + higher ECSW energy (0.16 mJ/mm^2^/120 impulses applied into the pelvic surface area at 3 h and days 3, 7, 14, 21 and 28 after ketamine administration)] and animals were euthanized by day 42 after ketamine administration.

#### 2.2.2. Urodynamic Test (i.e., Bladder Pressure Measurement)

The method for measuring the intravesical pressure (IVP) was based on our previous investigation [19]. Briefly, rats were anesthetized by two percent of inhalated isoflurane, followed by putting the animals in supine-position on a warming blanket that was maintained at 37 °C. A small catheter (PE50, Clay Adams, NJ, USA), which was advanced forward to the urethra, followed by entrance into the urinary bladder and then connected to a pressure transducer (BP Transducer Model MLT0380, Ad Instruments, Bella Vista, NSW, Australia) and syringe pump (Microinjection pump Model KDS100, KD Scientific Inc. 84 October Hill Road Holliston, MA 01756, USA) that transferred saline to the urinary bladder at a rate of 0.05 mL/min. The analog signals that were recorded continuously for 9 min, was converted into real-time and electrical signals (PowerLab 16/35 Model PL3516, Ad Instruments, Bella Vista, NSW, Australia), and amplified (Bridge Amp Model FE221, and Animal Bio Amp Model: FE136, Ad Instruments, Bella Vista, NSW, Australia) before being stored in a computer for further analysis (PowerLab 16/35 Model PL3516, Ad Instruments, Bella Vista, NSW, Australia).

#### 2.2.3. Isovolumetric Cystometrogram

The procedure and protocol were based on a previous report [20]. In detail, animals were anesthetized with 2.0% inhalational isoflurane. After emptying the bladder, the urethral catheter was placed to fill the bladder and infused with saline at a steady rate (0.08 mL/min) to measure the bladder pressure. Pressure and force signals were amplified (PowerLab 16/35 Model PL3516, Ad Instruments, Bella Vista, NSW, Australia, Bridge Amp Model FE221, and Animal Bio Amp Model: FE136, Ad Instruments, Bella Vista, NSW, Australia). Cystometrogram parameters, including the filling pressure, the peak micturition pressure, the frequency of non-voiding contractions (contraction without urine leakage during infusion) and the bladder capacity, were recorded.

#### 2.2.4. Collection of 18-h Urine for Assessment of Bladder-Maintained Maximal Urine Volume Prior to Micturition at Days 7, 14 and 28 after Ketamine Treatment

The procedure and protocol have been described in our previous report [13]. In detail, 18-h urine was collected in all animals at days 7, 14 and 28 after ketamine administration. For the collection of 18-h urine for the individual study, each animal was put in a metabolic cage [DXL-D, space: 190 × 290 × 550, Suzhou Fengshi Laboratory Animal Equipment Co. Ltd., Mainland China] for 18 h with free access to food and water.

### 2.3. Immunohistochemical (IHC) and Immunofluorescent (IF) Studies

The methods for IHC and IF examinations were based on our recent reports [13,14,15,16,17].

For quantification, three randomly selected high-power fields (HPFs, 200× for IHC and 400× for IF studies) were assessed in each section. The % of positively-stained cells per HPF for each animal was then analyzed, respectively. 

The integrated area (µm^2^) of fibrosis in each slice was analyzed by Image Tool 3 (IT3) image analysis software (University of Texas, Health Science Center, San Antonio, UTHSCSA; Image Tool for Windows, Version 3.0, USA). Three selected sections were analyzed for each animal. Three randomly selected HPFs (100×) were assessed in each section. After determining the number of pixels in each fibrotic area per HPF, the numbers of pixels obtained from three HPFs were summarized. The procedure was repeated in two other sections for each rat. The mean pixel number per HPF for each animal was then analyzed by summing up all pixel numbers and divided by nine. The mean integrated area (µm^2^) of fibrosis in quadriceps per HPF was obtained using a conversion factor of 19.24 (1 µm^2^ corresponded to 19.24 pixels).

### 2.4. Western Blot Analysis

The methods for Western blot measurements have been described in our recent studies [13,14,15,16,17]. Briefly, 50 μg of protein was extracted and then loaded and separated by SDS-PAGE using acrylamide gradients. After electrophoresis, the separated proteins were transferred electrophoretically to a PVDF membrane (Amersham Biosciences Inc. Buckinghamshire, UK). Nonspecific sites were blocked by incubation of the membrane in blocking buffer for 12 h. The membranes were incubated with the primary antibodies for 1 h at room temperature. Horseradish peroxidase-conjugated anti-rabbit immunoglobulin IgG (1:3000, Cell Signaling Technology, Inc., Danvers, MA, USA) was used as a secondary antibody for one-hour incubation at room temperature.

### 2.5. ELISA Assessment for Time Courses of Circulating Levels of Proinflammatory Cytokines

Circulatory levels of IL-6 and TNF-α, two inflammatory cytokines, were analyzed using duplicated determination with a commercially available ELISA method (R&D Systems, Minneapolis, MN, USA).

### 2.6. Statistical Analysis

Variables are expressed as mean ± SD. Statistical analysis was carried out using ANOVA followed by Bonferroni multiple-comparison post hoc test. SAS statistical software for Windows version 8.2 (SAS Institute, Cary, NC, USA) was utilized. A two-tailed probability with *p*-value < 0.05 was considered statistically significant.

## 3. Results

### 3.1. Impact of ECSW Therapy on Protecting the Rat Bladder Smooth Muscle Cells (RBdSMCs) against Oxidative Stress and Mitochondrial Damage

To elucidate whether the ECSW therapy would protect RBdSMCs against the oxidative-stress substance (i.e., menadione) damage, the cell culture was categorized into G1 (sham-control), G2 [RBdSMCs + menadione (25 μM)], G3 [RBdSMCs + ECSW (0.12 mJ/mm^2^, total 180 shots)] and G4 (RBdSMCs + menadione + ECSW), and Western blot was utilized. The result showed that the protein expressions of NOX-1, NOX-2 and oxidized protein, and three indicators of oxidative stress, were significantly increased more in G2 than in G1, G3 and G4, and significantly increased more in G4 than in G1 and G3, but they showed no difference between G1 and G3. Additionally, the flow cytometric analysis demonstrated that the fluorescent intensity of DCFH-DA, an indicator of total cellular oxidative stress, exhibited an identical pattern of oxidative stress in protein levels among the four groups (Figure 1).

Furthermore, the protein expression of cytosolic-cytochrome-C and cyclophilin-D, two indicators of mitochondrial-damaged biomarkers, displayed an identical pattern of oxidative stress among the four groups.

### 3.2. Impact of ECSW Therapy on Suppressing Menadione-Induced Inflammatory Reaction in HBdSMCs

Next, we wanted to test whether ECSW therapy would inhibit menadione induced inflammation by utilizing cell culture and Western blot. As we expected, the protein expressions of MyD88, TRAF6, p-IKB-α and NF-κB, four indicators of upstream inflammatory signaling, and protein expressions of TNF-α, IL-1ß, IL-6, MMP-9 and iNOS, five indices of downstream inflammatory signaling, were significantly increased in G2 moreso than in G1, G3 and G4, and significantly increased in G4 moreso than in G1 and G3, but they showed no difference between G1 and G3 (Figure 2).

### 3.3. Impact of ECSW Therapy on Regulating the Cell-Stress Signaling in HBdSMCs

The protein expressions of ASK1, p-MKK4, p-MKK7, ERK1/2, p-JNK, p-p38 and p-53, seven indices of cell-stress response signaling, were significantly increased moreso in G2 than in G1 and G3, and significantly reversed in G4 (all *p* < 0.0001) but they showed no difference between G1 and G3 (Figure 3).

### 3.4. Impact of ECSW Therapy on Attenuating the Menadione-Induced DNA Damage in HBdSMCs and Time Courses of Rat Urinary Level of Inflammatory Biomarkers

To assess whether ECSW therapy could ameliorate the oxidative-stress induced cell senescence and DNA damage, immunofluorescent stain was utilized in the present study. The result of microscopic findings demonstrated that the positively-stained γ-H2AX cells, an indicator of DNA damage, was significantly increased in G2 moreso than in G1, G3 and G4, and significantly increased in G4 moreso than in G1 and G3, but they showed no difference between G1 and G3 (Figure 4).

To measure whether ECSW applied to urinary bladder would reduce ketamine induced inflammatory reaction, the ELISA method was utilized to detect and quantify the serial changes of urinary soluble proinflammatory cytokines. The urinary levels of TNF-α and IL-6, two indices of proinflammatory cytokines, were highest in group 2 (i.e., ketamine treated only), lowest in group 1 (sham-operated control) and significantly higher in group 3 [ketamine + ECSW (0.12 mJ/mm^2^, 120 impulses, i.e., optimal ECSW energy)] than in group 4 [ketamine + ECSW (0.16 mJ/mm^2^, 120 impulses, i.e., higher ECSW energy)] not only by day 1 but also at days 7, 14 and 28 after ketamine treatment, suggesting that higher ECSW energy would perform better than the lower counterpart for preventing ketamine from damaging the urinary bladder (Figure 4).

### 3.5. Impact of ECSW on Inhibiting Ketamine-Induced Urine Frequency, Time Interval of Bladder Contraction and Bladder Maximal Pressure

To determine whether ECSW therapy could reduce the abnormal urination frequency, we measured 18 h-urination features of bladder. The result demonstrated that as compared with group 1, the time interval (i.e., duration) of urinary bladder contraction (i.e., an indicator of frequency of micturition) (Figure 5A,C) was significantly reduced and the maximal urinary bladder pressure (Figure 5B) was significantly increased (i.e., an indicator of difficulty in urinary bladder relaxation) in group 2. These findings were mimicked to the clinical setting of a patient who is a ketamine abuser with voiding difficulty. However, these phenomena were reversed in group 3 and even more reversed in group 4, suggesting that ECSW therapy effectively prevented ketamine induced bladder dysfunction (Figure 5).

### 3.6. Mean Bladder-Contained mMaximal Urine Volume and the Urinary Bladder Compliance Prior to Micturition by 18 h Urine Collection and the Urinary Bladder Weight by Day 42 after Ketamine Administration

To further realize the impact of ECSW on improving the capacity of urinary retaining maximal urine volume prior to micturition after ketamine treatment, we collected the 18-h urine amount and recorded time interval (i.e., duration) between urination in each group of animals by days 7, 14 and 28. Thus, the bladder-maintained maximal urine volume prior to micturition is equal to the total amount of urine accumulated divided by the number of micturition in the period of 18 h. The result showed that by days 7, 14 and 28, as compared with the group 1, the mean bladder-contained maximal urine volume prior to micturition was notably reduced in group 2, significantly progressively increased in group 3 and more significantly progressively increased in group 4, suggesting higher ECSW energy (i.e., 0.16 mJ/mm^2^) was better than the lower ECSW counterpart (i.e., 0.12 mJ/mm^2^) for maintaining the normal urine amount in bladder in condition of the ketamine-treated animals. Consistently, the urinary bladder compliance exhibited a similar pattern of maximal urine volume amount among the groups (Figure 6).

Additionally, by day 42, the pathologic analysis demonstrated that the morphological features of urinary bladder thickness/muscularization were notably higher in group 1 than in group 2, and were reversed in groups 3 and 4, whereas the bladder swelling exhibited an opposite pattern of bladder thickness among the four groups. Furthermore, the bladder weight was lowest in group 1, highest in group 2 and significantly increased in group 3 moreso than in group 4 (Figure 6).

### 3.7. ECSW Therapy Reduced the Ketamine-Induced Inflammatory Reaction, Fibrosis and Keratinization of Urinary Bladder by Day 42 after Ketamine Administration

The IF microscopic finding showed that the cellular expressions of COX-2 and substance p (Figure 7), two indices of cellular level of inflammation, were lowest in group 1, highest in group 2 and significantly lower in group 4 than in group 3. Additionally, the IHC stain revealed that CK18 (Figure 8), a keratinized marker in the epithelial layer of the urinary bladder, exhibited an identical pattern of inflammation among the four groups. Moreover, the Masson’s trichrome stain identified that the fibrosis area (Figure 8) in urinary bladder muscle also exhibited an identical pattern of inflammation among the four groups (Figure 7 and Figure 8).

### 3.8. ECSW Suppressed the Protein Levels of Oxidative Stress, Apoptosis, Fibrosis and Mitochondrial Damage in Rat Urinary Bladder by Day 42 after Ketamine Administration

The protein expressions of NOX-1, NOX-2 and oxidized protein, three indicators of oxidative stress, were lowest in group 1, highest in group 2 and significantly lower in group 4 than in group 3. Additionally, the protein expressions of cleaved caspase 3, cleaved PARP and mitochondrial Bax, three indicators of apoptosis, and protein expressions of Smad3 and TGF-ß, two indices of fibrosis, displayed an identical pattern of oxidative stress among the four groups. Furthermore, the protein expressions of cyclophilin D and cytosolic cytochrome-C, two indicators of mitochondrial damage biomarkers, also exhibited an identical pattern of oxidative stress among the four groups (Figure 9).

### 3.9. ECSW Suppressed the Protein Levels of Inflammatory Signalings in Rat Urinary Bladder by Day 42 after Ketamine Administration

The protein expressions of TLR-2, TLR-4, MyD88, TRAF6, p-IKB-α and NF-κB, six indices of upstream inflammatory signaling, and protein expressions of TNF-α, IL-1ß, MMP-9, COX-2 and iNOS, five indices of downstream inflammatory signaling, were lowest in group 1, highest in group 2 and significantly lower in group 4 than in group 3 (Figure 10).

### 3.10. Impact of ECSW Therapy on Regulating the Protein Levels of Cell-Stress Response Signalings in Rat Urinary Bladder by Day 28 after Ketamine Administration

The protein expressions of ASK1, p-MKK4, p-MKK7, ERK1/2, p-JNK, and p-p38, six indicators of cell-stress response signalings, were highest in group 2, lowest in group 1 and significantly higher in group 3 than in group 4, suggesting these were intrinsic responses to ketamine stimulation that were downregulated by ECSW therapy (Figure 11).

## 4. Discussion

This study, which investigated the therapeutic impact of ECSW against ketamine induced rat urinary bladder dysfunction, yielded several striking preclinical implications. First, the study not only found the ECSW therapy to be effective, but it also clarified that the higher ECSW energy was more effective than the lower energy counterpart (i.e., defined as optimal energy) in protecting the functional and architectural integrity of the urinary bladder. Second, this study delineated that ECSW therapy on preserving the functional and architectural integrity of the urinary bladder was mainly through regulating the oxidative-stress, inflammatory and cell-stress signaling pathways.

Abundant data have shown that damage to the organs always elicits [13,14,15,16,17,18,19] an inflammatory reaction and the generation of oxidative stress. Interestingly, our previous study has demonstrated that ECSW therapy effectively protected cyclophosphamide-induced acute cystitis in rodents mainly through inhibiting inflammation and oxidative stress [13]. Based on these findings [13,14,15,16,17,18,19], by utilization of the rat bladder smooth muscle cell line (i.e., CSC-C9375W), our in vitro study aimed to elucidate the relevant signaling upregulated by oxidative-stress compound (i.e., menadione). In this way, several remarkable molecular signaling pathways were searched and further identified. First, menadione treatment markedly enhanced the protein expressions of oxidative stress, which in turn caused protein expressions of mitochondrial damage (i.e., upregulated cytosolic cytochrome C and cyclophilin D) (refer to Figure 1). Second, menadione treatment significantly augmented upstream and downstream inflammatory signalings (refer to Figure 2). Third, menadione treatment also significantly upregulated cell stress response signaling (refer to Figure 3). Based on the findings of the previous studies [13,14,15,16,17,18,19] and results (Figure 1, Figure 2 and Figure 3) of our in vitro study, we therefore performed the animal study undergoing ketamine-induced urinary bladder dysfunction and ECSW treatment.

An essential finding of our animal model study was that, as compared to the SC group, the maximal bladder-reserved urine volume in the urine bladder just prior to micturition, i.e., an index of bladder functional integrity, was substantially reduced in ketamine-treated animals (refer to Figure 7). Additionally, another three indices of bladder functional integrity, including the interval of bladder contraction and the duration of micturition were significantly longer and bladder pressure was significantly reduced in the SC group than those in the ketamine-treated group (refer to Figure 6). One important finding was that these parameters were significantly reversed by lower energy (i.e., 0.12 mJ/mm^2^) and more significantly reversed by higher energy (i.e., 0.16 mJ/mm^2^) of ECSW therapy. We suggest that our finding is full of novelty not only due to the fact that this was the first reported study to utilize ECSW therapy for ketamine-induced urinary bladder dysfunction, but also but also because we found that such strategic management was attractive and promising. In this way, we recommend that perhaps such a modality could be applied to those ketamine-abusers who have severe urinary bladder dysfunction and are refractory to conventional therapy.

Another essential finding was that when we scrutinized the anatomic and pathological findings, we found that the bladder weight, the keratinized epithelial layer and fibrotic area of the bladder were more markedly increased in ketamine-treated animals than in those of SC animals. These novel findings highlight that ketamine abuse not only damaged the urinary bladder function but also destroyed the anatomical and architectural integrity of the urinary bladder. However, this anatomical and ultrastructural integrity of the bladder was notably preserved by lower energy and more preserved by higher energy of ECSW.

The link between inflammatory reaction and generation of oxidative stress and organ damage has been extensively investigated by previous studies [13,14,15,16,17,18,19,20,21,22]. One important finding in the present study was that the upstream and downstream inflammatory signalings along with the cellular level of inflammation were significantly increased in ketamine-treated animals than in those of SC animals. Consistently, the oxidative stress signaling was also upregulated in the ketamine-treated group moreso than in the SC group. Our findings, in addition to corroborating the findings of previous studies [13,14,15,16,17,18,19,20,21,22], could mainly explain why the urinary bladder function and anatomical structure were significantly destroyed in ketamine-treated animals. Similarly, they could also explain why the protein levels of apoptosis and fibrosis were significantly increased in ketamine-treated rodents as opposed to SC rodents. However, these activated inflammatory and oxidative-stress signalings, as well as apoptotic and fibrotic biomarkers, were substantially suppressed by lower energy and more substantially suppressed by high energy of ECSW.

Our previous studies have clearly identified that cell stress signaling is frequently activated by ischemic and hypoperfusion conditions [23,24]. A principal finding in the present study was that the cell stress signaling in response to ketamine stimulation was remarkably increased in ketamine treated animals moreso than in SC animals. Our finding, in addition to supporting the findings of previous studies [23,24], suggests that it was an intrinsically struggling response of the urinary bladder smooth muscle cells to ketamine stimulation with a distinctive feature being that as more intensive stimulation was applied, the stronger the cell signaling was in response. However, this cell stress signaling was significantly attenuated in lower energy and more significantly attenuated in higher energy of ECSW.

## 5. Study Limitation

This study has limitations. First, although the study period was 42 days, it remained too short to reflect the long-term outcomes of ketamine treatment with and without ECSW therapy. Second, ketamine induced bladder dysfunction varies widely from totally recoverable to a seriously contracted bladder. Thus, relatively short-term (i.e., 42 days study period) ketamine treatment might affect the bladder merely to a certain degree that could be reversible, even without medication. Third, stepwise increased energy of ECSW did not perform in the present study. Accordingly, we did not know whether a further higher ECSW energy would provide more promising outcomes for the ketamine-treated rodent. 

In conclusion, the results of the present study provided adequate evidence that ECSW therapy promisingly protected the functional and structural integrity against bladder ketamine damage.

## Figures and Tables

**Figure 1 biomedicines-09-01391-f001:**
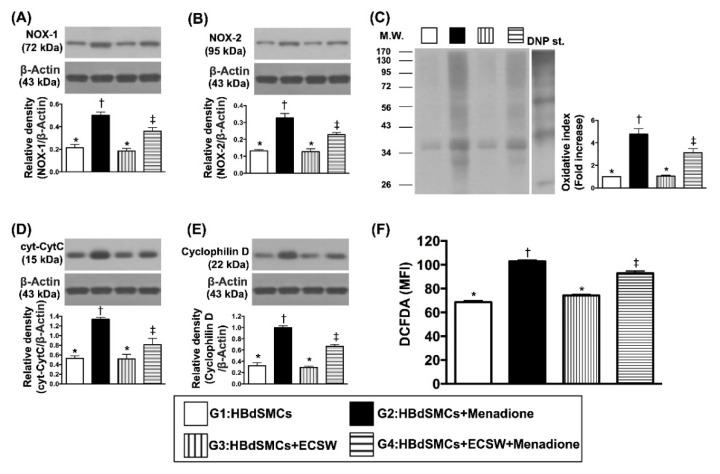
ECSW therapy protected the rat bladder smooth muscle cells (RBdSMCs) against oxidative stress and mitochondrial damage. (**A**) Protein expression of NXO-1, * vs. other groups with different symbols (†, ‡), *p* < 0.001. (**B**) Protein expression of NOX-2, * vs. other groups with different symbols (†, ‡), *p* < 0.001. (**C**) The oxidized protein expression, * vs. other groups with different symbols (†, ‡), *p* < 0.001 (Note: The left and right lanes shown on the upper panel represent protein molecular weight marker and control oxidized molecular protein standard, respectively). M.W. = molecular weight; DNP = 1–3 dinitrophenylhydrazone. (**D**) Protein expression of cytosolic cytochrome C (cyt-CytC), * vs. other groups with different symbols (†, ‡), *p* < 0.001. (**E**) Protein expression of cyclophilin D (cyc-D), * vs. other groups with different symbols (†, ‡), *p* < 0.001. (**F**) Flow cytometric analysis of mean fluorescent intensity (MFI) of DCFDA [i.e., reactive oxygen species (ROS)], * vs. other groups with different symbols (†, ‡), *p* < 0.001. All statistical analyses were performed by one-way ANOVA, followed by Bonferroni multiple comparison post hoc test (*n* = 6 for each group). Symbols (*, †, ‡) indicate significance (at 0.05 level). ECSW = extracorporeal shock wave; RBdSMCs = rat bladder smooth muscle cells.

**Figure 2 biomedicines-09-01391-f002:**
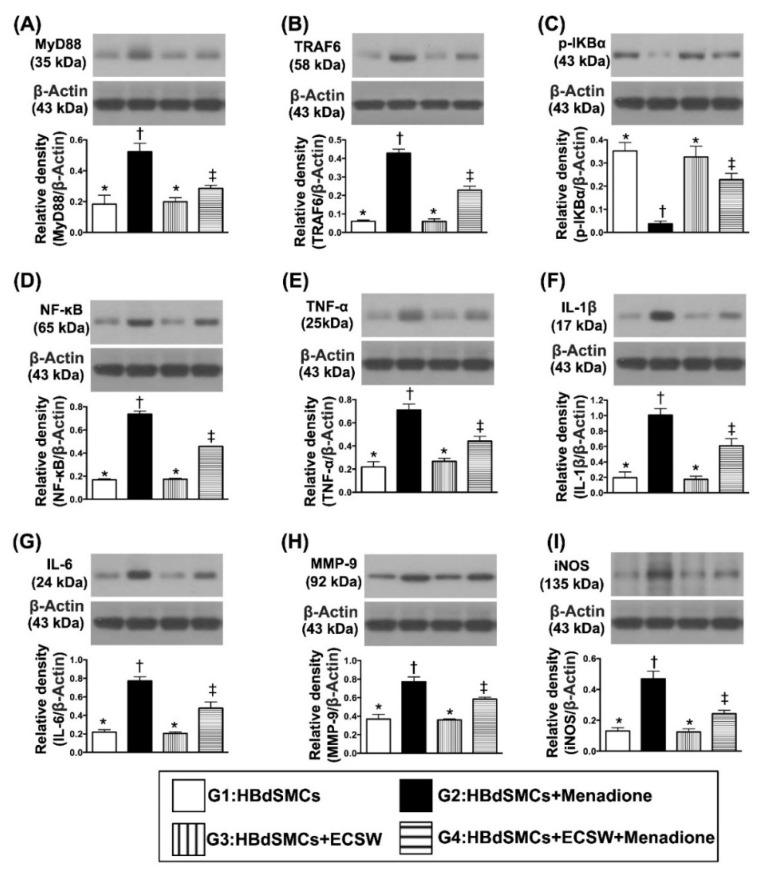
ECSW therapy markedly suppressed menadione-induced inflammatory reaction in RBdSMCs. (**A**) Protein expression of myeloid differentiation primary response 88 (MyD88), MyD88, * vs. other groups with different symbols (†, ‡), *p* < 0.001. (**B**) Protein expression of TNF receptor associated factor 6 (TRAF6), * vs. other groups with different symbols (†, ‡), *p* < 0.001. (**C**) Protein expression of phosphorylated (p)-IKB-α, * vs. other groups with different symbols (†, ‡), *p* < 0.001. (**D**) Protein expression of nuclear factor-κB (NF-κB), * vs. other groups with different symbols (†, ‡), *p* < 0.001. (**E**) Protein expression of phosphorylated tumor necrosis factor alpha (TNF-α), * vs. other groups with different symbols (†, ‡), *p* < 0.001. (**F**) Protein expression of interleukin (IL)-1ß, * vs. other groups with different symbols (†, ‡), *p* < 0.001. (**G**) Protein expression of IL-6, * vs. other groups with different symbols (†, ‡), *p* < 0.001. (**H**) Protein expression of matrix metalloproteinase 9 (MMP-9), * vs. other groups with different symbols (†, ‡), *p* < 0.001. (**I**) Protein expression of induced nitric oxide synthase (iNOS), * vs. other groups with different symbols (†, ‡), *p* < 0.001. All statistical analyses were performed by one-way ANOVA, followed by Bonferroni multiple comparison post hoc test (*n* = 6 for each group). Symbols (*, †, ‡) indicate significance (at 0.05 level). ECSW = extracorporeal shock wave; RBdSMCs = rat bladder smooth muscle cells.

**Figure 3 biomedicines-09-01391-f003:**
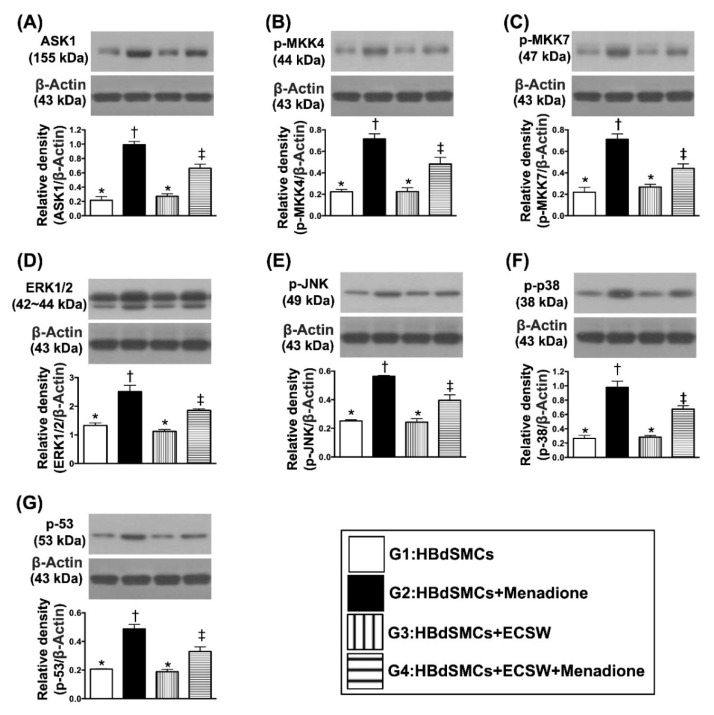
ECSW therapy regulated the cell-stress signaling in RBdSMCs. (**A**) Protein expression of Apoptosis signal-regulating kinase 1 (ASK1), * vs. other groups with different symbols (†, ‡), *p* < 0.001. (**B**) Protein expression of phosphorylated Mitogen-activated protein kinase kinase 4 (p-MKK4), * vs. other groups with different symbols (†, ‡), *p* < 0.001. (**C**) Protein expression of p-MKK7, * vs. other groups with different symbols (†, ‡), *p* < 0.001. (**D**) Protein expression of ERK1/2, * vs. other groups with different symbols (†, ‡), *p* < 0.001. (**E**) Protein expression of phosphorylated c-Jun N-terminal kinases (p-JNK), * vs. other groups with different symbols (†, ‡), *p* < 0.001. (**F**) Protein expression of p-p38, * vs. other groups with different symbols (†, ‡), *p* < 0.001. (**G**) Protein expression of p-53, * vs. other groups with different symbols (†, ‡), *p* < 0.001. All statistical analyses were performed by one-way ANOVA, followed by Bonferroni multiple comparison post hoc test (*n* = 6 for each group). Symbols (*, †, ‡) indicate significance (at 0.05 level). ECSW = extracorporeal shock wave; RBdSMCs = rat bladder smooth muscle cells.

**Figure 4 biomedicines-09-01391-f004:**
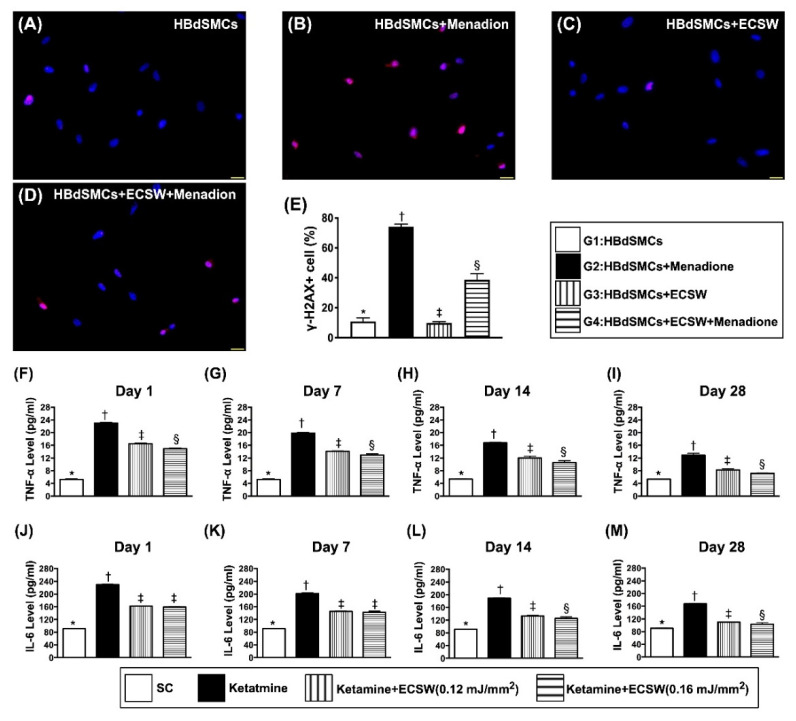
ECSW therapy attenuated menadione-induced DNA damage in RBdSMCs and time courses of urinary level of inflammatory biomarkers. (**A**–**D**) Illustrating the immunofluorescent microscopic finding (400×) for identification of positively stained γ-H2AX cells (pink color). (**E**) Analytical result of number of γ-H2AX+ cells, * vs. other groups with different symbols (†, ‡, §), *p* < 0.0001. Scale bar in right lower corner represents 20 µm. (**F**–**I**) The urine levels of tumor necrosis factor (TNF)-α by days 1 (**F**), 7 (**G**), 14 (**H**) and 28 (**I**), respectively, analytical result of TNF-α in urine, * vs. other groups with different symbols (†, ‡, §), *p* < 0.0001. (**J**–**M**) The urine levels of interleukin (IL)-6 by days 1 (**J**), 7 (**K**), 14 (**L**) and 28 (**M**), respectively, analytical result of interleukin (IL)-6 in urine, analytical result of IL-6 in urine, for days 1 and 7, * vs. other groups with different symbols (†, ‡), *p* < 0.0001; for days 14 and 28, * vs. other groups with different symbols (†, ‡, §), *p* < 0.0001. All statistical analyses were performed by one-way ANOVA, followed by Bonferroni multiple comparison post hoc test (*n* = 6 for each group). Symbols (*, †, ‡) indicate significance (at 0.05 level). ECSW = extracorporeal shock wave; RBdSMCs = rat bladder smooth muscle cells.

**Figure 5 biomedicines-09-01391-f005:**
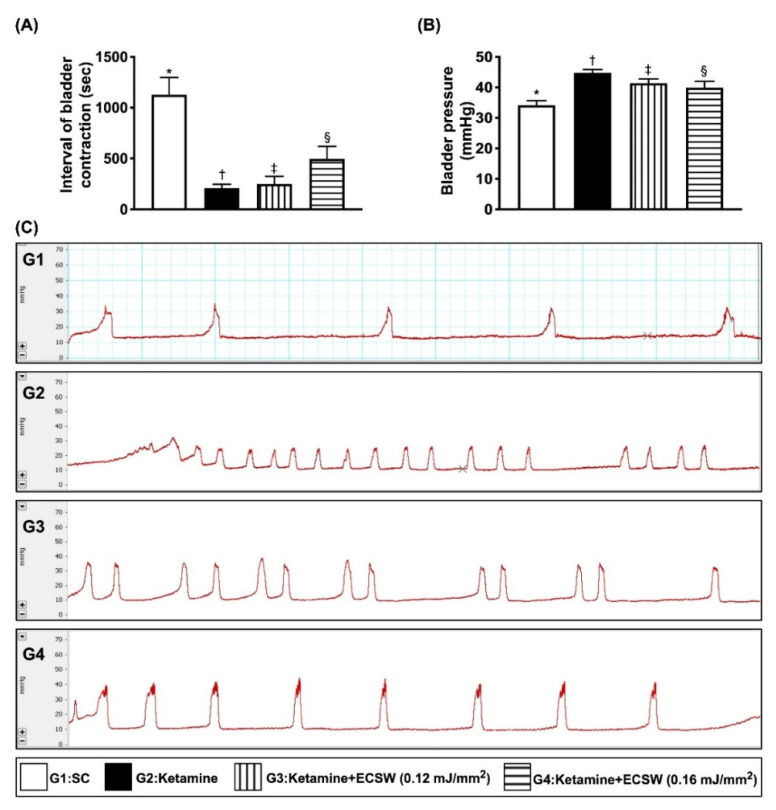
ECSW therapy inhibited ketamine-induced urine frequency, time interval of bladder contraction and bladder maximal pressure. (**A**) The time interval of urinary bladder contraction, * vs. other groups with different symbols (†, ‡, §), *p* < 0.0001. (**B**) Maximal urinary bladder pressure, * vs. other groups with different symbols (†, ‡, §), *p* < 0.0001. (**C**) Illustrating the time interval of urinary bladder contraction (i.e., the frequency) among the four groups. The frequency of urinary bladder contraction in G2 was remarkably increased as compared with G3 and G4 and more remarkably increased as compared with G1 (i.e., sham-control). All statistical analyses were performed by one-way ANOVA, followed by Bonferroni multiple comparison post hoc test (*n* = 6 for each group). Symbols (*, †, ‡, §) indicate significance (at 0.05 level). ECSW = extracorporeal shock wave.

**Figure 6 biomedicines-09-01391-f006:**
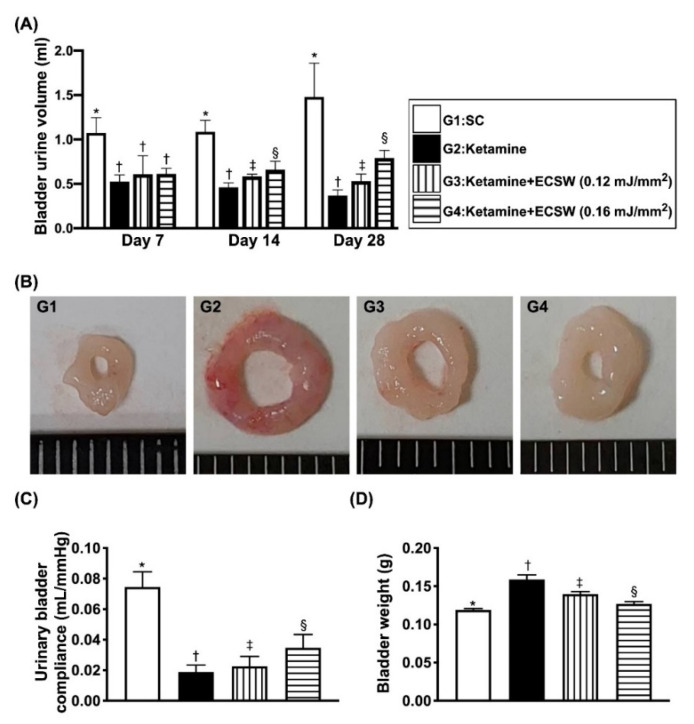
Time courses of mean maximal bladder urine volume per time interval prior to micturition (i.e., within 18 h collection of urine divided by total number of urination) and urinary bladder compliance, and the urinary bladder weight by day 42 after ketamine administration. (**A**) By day 7, the mean of urine volume prior to micturition, * vs. †, *p* < 0.001; by day 14, the mean of urine volume prior to micturition, * vs. other groups with different symbols (†, ‡, §), *p* < 0.0001; by day 28, the mean of urine volume prior to micturition, * vs. other groups with different symbols (†, ‡, §), *p* < 0.0001. (**B**) Illustrating the anatomical features of urinary bladder by day 42 after ketamine treatment. The appearance of urinary bladder was remarkably enlarged and swelling in G2 moreso than in G3 and G4 and more remarkably enlarged and swelling than in G1. (**C**) By day 28, the urinary bladder compliance, * vs. other groups with different symbols (†, ‡, §), *p* < 0.0001. (**D**) Analytical result of bladder weight, * vs. other groups with different symbols (†, ‡, §), *p* < 0.001. All statistical analyses were performed by one-way ANOVA, followed by Bonferroni multiple comparison post hoc test (*n* = 6 for each group). Symbols (*, †, ‡, §) indicate significance (at 0.05 level). ECSW = extracorporeal shock wave.

**Figure 7 biomedicines-09-01391-f007:**
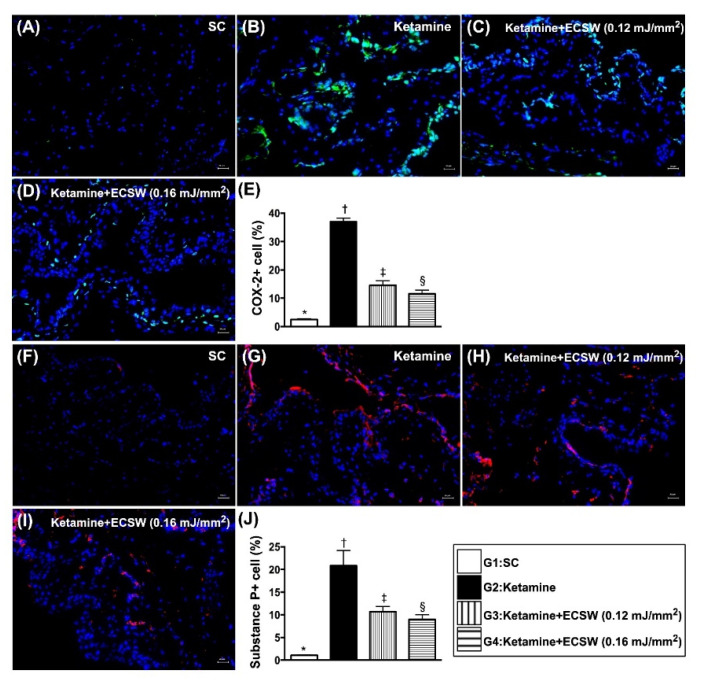
ECSW therapy reduced the ketamine-induced inflammatory cell infiltration in rat urinary bladder by day 42 after ketamine administration. (**A**–**D**) Illustrating the immunofluorescent (IF) microscopic finding (400×) for identification of positively-stained COX-2 cells (green color). (**E**) Analytical result of percentage of COX-2+ cells in high-power field, * vs. other groups with different symbols (†, ‡, §), *p* < 0.0001. (**F**–**I**) Illustrating the IF microscopic finding (400×) for identification of positively-stained substance P cells (red color). (**J**) Analytical result of percentage of substance P+ cells in high-power field, * vs. other groups with different symbols (†, ‡, §), *p* < 0.0001. Scale bar in right lower corner represents 20 µm. All statistical analyses were performed by one-way ANOVA, followed by Bonferroni multiple comparison post hoc test (*n* = 6 for each group). Symbols (*, †, ‡, §) indicate significance (at 0.05 level). ECSW = extracorporeal shock wave.

**Figure 8 biomedicines-09-01391-f008:**
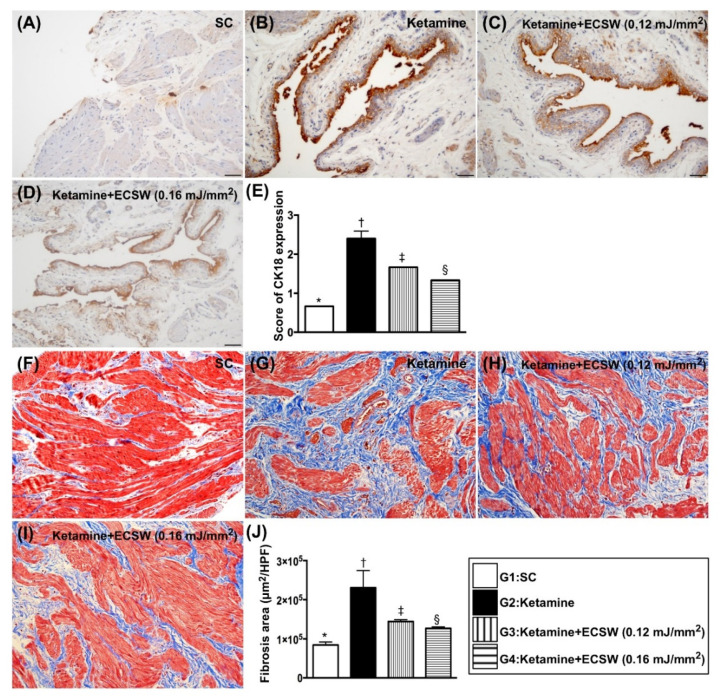
ECSW therapy reduced the ketamine-induced fibrosis and keratinization of urinary bladder by day 42 after ketamine administration. (**A**–**D**) Illustrating the immunohistochemical (IHC) microscopic finding (200×) for identification of IHC stained intensity of CK18 in urinary bladder epithelial layer (gray color). (**E**) Scoring intensity of CK18 positively-stained expression, * vs. other groups with different symbols (†, ‡, §), *p* < 0.0001. (**F**–**I**) Illustrating the Masson’s trichrome stain (200×) for identification of fibrosis area in urinary bladder muscle (blue color). (**J**) Analytical result of fibrotic area, * vs. other groups with different symbols (†, ‡, §), *p* < 0.0001. Scale bar in right lower corner represents 50 µm. All statistical analyses were performed by one-way ANOVA, followed by Bonferroni multiple comparison post hoc test (*n* = 6 for each group). Symbols (*, †, ‡, §) indicate significance (at 0.05 level). ECSW = extracorporeal shock wave.

**Figure 9 biomedicines-09-01391-f009:**
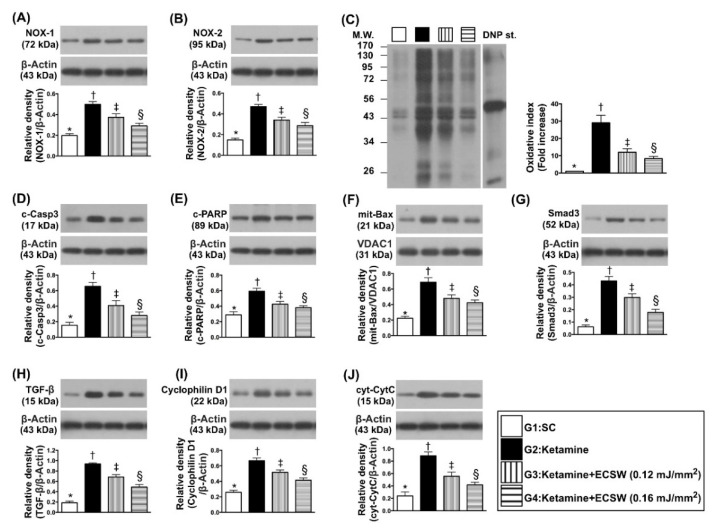
ECSW suppressed the protein levels of oxidative stress, apoptosis, fibrosis and mitochondrial damage in rat urinary bladder by day 42 after ketamine administration. (**A**) Protein expression of NOX-1, * vs. other groups with different symbols (†, ‡, §), *p* < 0.0001. (**B**) Protein expression of NOX-2, * vs. other groups with different symbols (†, ‡, §), *p* < 0.0001. (**C**) The oxidized protein expression, * vs. other groups with different symbols (†, ‡, §), *p* < 0.0001 (Note: The left and right lanes shown on the upper panel represent protein molecular weight marker and control oxidized molecular protein standard, respectively). M.W. = molecular weight; DNP = 1–3 dinitrophenylhydrazone. (**D**) Protein expression of cleaved caspase 3 (c-Casp3), * vs. other groups with different symbols (†, ‡, §), *p* < 0.0001. (**E**) Protein expression of cleaved Poly ADP-ribose polymerase (c-PARP), * vs. other groups with different symbols (†, ‡, §), *p* < 0.0001. (**F**) Protein expression of mitochondrial (mit)-Bax, * vs. other groups with different symbols (†, ‡, §), *p* < 0.0001. (**G**) Protein expression of Smad3, * vs. other groups with different symbols (†, ‡, §), *p* < 0.0001. (**H**) Protein expression of transforming growth factor (TGF)-ß, * vs. other groups with different symbols (†, ‡, §), *p* < 0.0001. (**I**) Protein expression of cyclophilin D, * vs. other groups with different symbols (†, ‡, §), *p* < 0.0001. (**J**) Protein expression of cytosolic cytochrome c (cyt-CytC), * vs. other groups with different symbols (†, ‡, §), *p* < 0.0001. All statistical analyses were performed by one-way ANOVA, followed by Bonferroni multiple comparison post hoc test (*n* = 6 for each group). Symbols (*, †, ‡, §) indicate significance (at 0.05 level). ECSW = extracorporeal shock wave.

**Figure 10 biomedicines-09-01391-f010:**
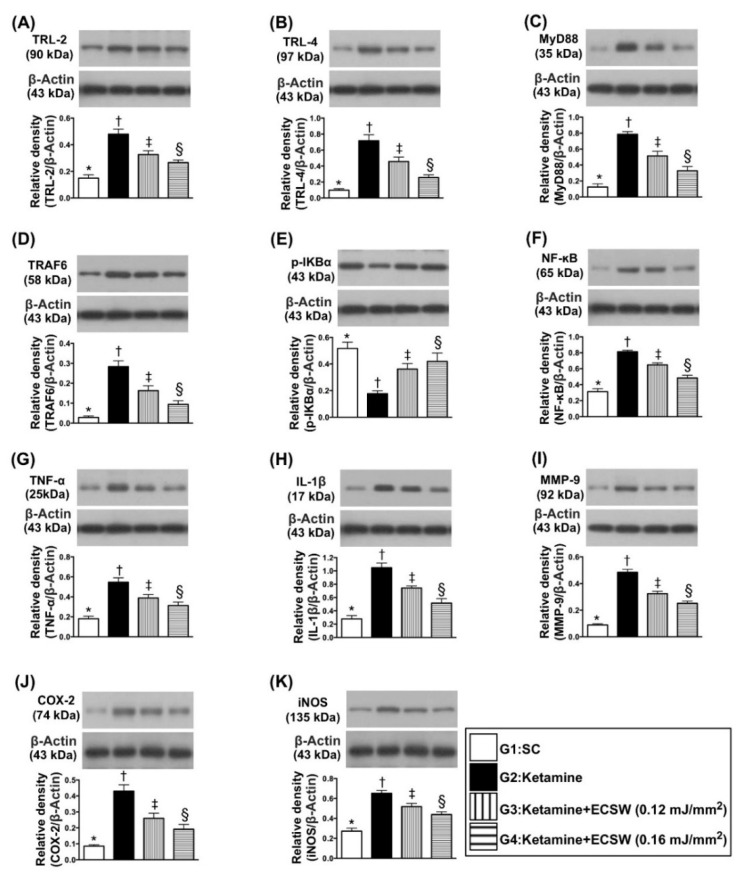
ECSW therapy suppressed the protein levels of inflammatory signalings in rat urinary bladder by day 42 after ketamine administration. (**A**) Protein expression of Toll-like receptor (TLR)-2, * vs. other groups with different symbols (†, ‡, §), *p* < 0.0001. (**B**) Protein expression of TLR-4, * vs. other groups with different symbols (†, ‡, §), *p* < 0.0001. (**C**) Protein expression of MyD88, * vs. other groups with different symbols (†, ‡, §), *p* < 0.0001. (**D**) Protein expression of TRAF6, * vs. other groups with different symbols (†, ‡, §), *p* < 0.0001. (**E**) Protein expression of p-IKB-α, * vs. other groups with different symbols (†, ‡, §), *p* < 0.0001. (**F**) Protein expression of NF-κB, * vs. other groups with different symbols (†, ‡, §), *p* < 0.0001. (**G**) Protein expression of TNF-α, * vs. other groups with different symbols (†, ‡, §), *p* < 0.0001. (**H**) Protein expression of interleukin (IL)-1ß, * vs. other groups with different symbols (†, ‡, §), *p* < 0.0001. (**I**) Protein expression of MMP-9, * vs. other groups with different symbols (†, ‡, §), *p* < 0.0001. (**J**) Protein expression of COX-2, * vs. other groups with different symbols (†, ‡, §), *p* < 0.0001. (**K**) Protein expression of and iNOS, * vs. other groups with different symbols (†, ‡, §), *p* < 0.0001. All statistical analyses were performed by one-way ANOVA, followed by Bonferroni multiple comparison post hoc test (*n* = 6 for each group). Symbols (*, †, ‡, §) indicate significance (at 0.05 level). ECSW = extracorporeal shock wave.

**Figure 11 biomedicines-09-01391-f011:**
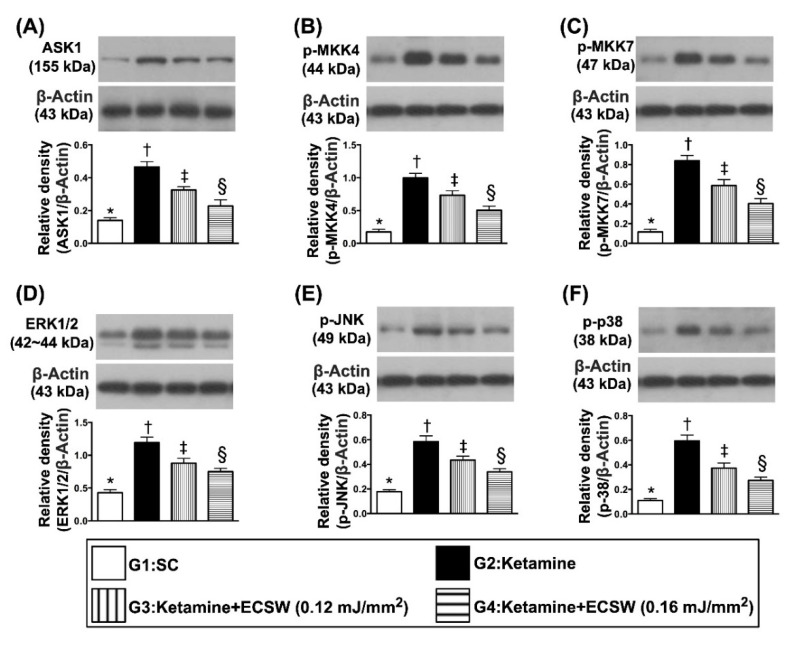
ECSW therapy regulated the protein levels of cell-stress response signalings in rat urinary bladder by day 42 after ketamine administration. (**A**) Protein expression of ASK1, * vs. other groups with different symbols (†, ‡, §), *p* < 0.0001. (**B**) Protein expression of p-MKK4, * vs. other groups with different symbols (†, ‡, §), *p* < 0.0001. (**C**) Protein expression of p-MKK7, * vs. other groups with different symbols (†, ‡, §), *p* < 0.0001. (**D**) Protein expression of ERK1/2, * vs. other groups with different symbols (†, ‡, §), *p* < 0.0001. (**E**) Protein expression of p-JNK, * vs. other groups with different symbols (†, ‡, §), *p* < 0.0001. (**F**) Protein expression of p-p38, * vs. other groups with different symbols (†, ‡, §), *p* < 0.0001. All statistical analyses were performed by one-way ANOVA, followed by Bonferroni multiple comparison post hoc test (*n* = 6 for each group). Symbols (*, †, ‡, §) indicate significance (at 0.05 level). ECSW = extracorporeal shock wave.

## Data Availability

The datasets of present study can be available from the corresponding author upon request.

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
