# Peer review of "Extracorporeal Shock Wave Therapy Protected the Functional and Architectural Integrity of Rodent Urinary Bladder against Ketamine-Induced Damage"

_biomedicines, 2021, doi:10.3390/biomedicines9101391_

Round 1
Reviewer 1 Report
This manuscript reported the molecular evidences for the therapeutic efficacy of low energy shock wave (LESW) in treatment of ketamine induced bladder dysfunctions in rats. The study design and experiment performed are well described and the results are clearly shown to provide evidence for their final conclusions. There are several drawbacks that authors should revise in their second draft of manuscript.
1) The aim of this study is to prove the effect of LESW on ketamine induced cystitis. The first paragraph of introduction is irrelevant to this theme. Please remove it and replace by a more thorough introduction of the effect of ketamine on bladder function.
2) Ketamine induced bladder dysfunction varies widely from totally recoverable to a severely contracted bladder. Short-term ketamine treatment could affect the bladder only to a degree that is reversible, even without medication. This point should be addressed in the discussion or limitation of the study.
3) One of the irreversible damaged bladder after different type of bladder injury is the deposition of collagen between detrusor muscle bundles. Although LESW seems effective in restoration of structural alignment and reverse the stress oxidative damage in ketamine cystitis, A severely inflammed and contracted bladder might not reversed by this treatment. Have the authors exam the collagen content between different subgroup of bladders?
4) In addition to the bladder contractility intervals and amplitude, bladder compliance is also an important parameter for evaluating bladder inflammation. Please add this parameter in the assessment item after LESW treatment in different subgroup.
Author Response
Dear Reviewer:
Your constructive criticism is greatly appreciated. We have made the following responses to comply with your honorable suggestions (Note: The revised parts of the manuscript in response to Reviewer’s comments have been marked in red color):
Response to Comments and Suggestions for Authors
Comment 1: The aim of this study is to prove the effect of LESW on ketamine induced cystitis. The first paragraph of introduction is irrelevant to this theme. Please remove it and replace by a more thorough introduction of the effect of ketamine on bladder function.
Response 1: Yes, according to your recommendation, we have rewritten the Introduction paragraph in our revised manuscript.
Comment 2: Ketamine induced bladder dysfunction varies widely from totally recoverable to a severely contracted bladder. Short-term ketamine treatment could affect the bladder only to a degree that is reversible, even without medication. This point should be addressed in the discussion or limitation of the study.
Response 2: Yes, according to your recommendation, we have discussed our study limitation in the Limitation Section of our revised manuscript.
Comment 3: One of the irreversible damaged bladder after different type of bladder injury is the deposition of collagen between detrusor muscle bundles. Although LESW seems effective in restoration of structural alignment and reverse the stress oxidative damage in ketamine cystitis, A severely inflammed and contracted bladder might not reversed by this treatment. Have the authors exam the collagen content between different subgroup of bladders?
Response 3: We really agree the reviewer’s comment and we are honest to tell you that we did not perform the more detail in separating the subgroup presentation of collagen fibers, such as type I, type II, type IV, etc. We apology for that we cannot repeated these data because as you know that the bladder specimen in a rat is always so small. Those harvested urinary bladder specimen had been exhausted due extensive works had been performed in the present study.
Comment 4: In addition to the bladder contractility intervals and amplitude, bladder compliance is also an important parameter for evaluating bladder inflammation. Please add this parameter in the assessment item after LESW treatment in different subgroup.
Response 4: As we know that the compliance = the change in volume (ΔV) divided by the change in pressure (ΔP). So, we have recalculated the data and the data of compliance has been added in Figure 6 of our revised manuscript.
We would like to take this opportunity to express our appreciation for your detailed review of the article and the kindness of giving us valuable suggestions. Thank you very, very much!
Reviewer 2 Report
This is a stringent paper exploring whether extracorporeal-shock-wave (ECSW) can protect the functional and anatomical integrity of rat urinary-bladder against ketamine-induced damage. The paper should be accepted after only some minor technical interventions, as I do not see any inconsistensies in their methodological approach and the results.
When stating intra-observer variability and intra-assay coefficients of variance, were they calculated from the calculated concentrations or the raw optical densities? In the Statistical analysis part of the manuscript, it should be emphasized that the probability value <0.05 is two-tailed, and also the abbreviation should be introduced (p-value).
Please include other studies published recently that are relevant to the topic in the Introduction or Discussion sections of the manuscript. One example: Yeh CH et al. Intravesical Instillation of Norketamine, a Ketamine Metabolite, and Induced Bladder Functional Changes in Rats. Toxics. 2021;9(7):154. doi: 10.3390/toxics9070154.
Author Response
Dear Reviewer:
Your constructive criticism is greatly appreciated. We have made the following responses to comply with your honorable suggestions (Note: The revised parts of the manuscript in response to Reviewer’s comments have been marked in pink color):
Response to reviewer’s comments to author
Comment 1: This is a stringent paper exploring whether extracorporeal-shock-wave (ECSW) can protect the functional and anatomical integrity of rat urinary-bladder against ketamine-induced damage. The paper should be accepted after only some minor technical interventions, as I do not see any inconsistensies in their methodological approach and the results.
Response 1: We sincerely thank you for your appreciation of our study.
Comment 2: When stating intra-observer variability and intra-assay coefficients of variance, were they calculated from the calculated concentrations or the raw optical densities? In the Statistical analysis part of the manuscript, it should be emphasized that the probability value <0.05 is two-tailed, and also the abbreviation should be introduced (p-value).
Response 2: (1) We apology for our inappropriate description in ELISA analysis. The sentence “Intra-observer variability of the measurements was also assessed, and the mean intra-assay coefficients of variance were all <4.5%.” were removed in our revised manuscript. (2) Yes, your suggestion has been added in the Statistical analysis part of our revised manuscript.
Comment 3: Please include other studies published recently that are relevant to the topic in the Introduction or Discussion sections of the manuscript. One example: Yeh CH et al. Intravesical Instillation of Norketamine, a Ketamine Metabolite, and Induced Bladder Functional Changes in Rats. Toxics. 2021;9(7):154. doi: 10.3390/toxics9070154.
Response 3: Thank you for your kindly suggestion. Yes, we have added this relevant paper in the Introduction paragraph of our revised manuscript.
We are greatly indebted to you for your professional comments.
Round 2
Reviewer 1 Report
The authors have appropriately revised this manuscript and added the required data in the results section. This manuscript is currently acceptable for publication in this journal.
This manuscript is a resubmission of an earlier submission. The following is a list of the peer review reports and author responses from that submission.